# Degradation of White Birch Shelterbelts by the Attack of White-Spotted Longicorn Beetles in Central Hokkaido, Northern Japan

**Kazuhiko Masaka [1,*], Yohichi Wakita [2], Kenta Iwasaki [2] and Masato Hayamizu [3]**

[1]  Department of Forest Science, Faculty of Agriculture, Iwate University, Morioka 020 8550, Japan

[2]  Doto Station, Forestry Research Institute, Hokkaido Research Organization, Shintoku 081 0038, Japan; wakita-youichi@hro.or.jp (Y.W.); iwasaki-kenta@hro.or.jp (K.I.)

[3]  Forestry Research Institute, Hokkaido Research Organization, Koshunai, Bibai 079 0198, Japan; hayamizu-masato@hro.or.jp

*   Correspondence: masaka@iwate-u.ac.jp

**Abstract:** A widespread decline of white birch (*Betula platyphylla* var. *japonica*) shelterbelts was observed in central Hokkaido, Japan. Many exit holes bored by white-spotted longicorn beetles (*Anoplophora malasiaca*) were found at the base of the trunks of trees in these stands. The present study aims to evaluate the effects of infestation on the degradation, and demonstrates whether the number of exit holes ($N_{holes}$) can be used as an index of the decline of trees. We selected 35 healthy appearing stands and 16 degraded stands in the study area. A generalized linear mixed model with zero inflation revealed that $N_{holes}$ of standing dead trees tended to be greater than that of living trees, and the tree vigor decreased with increasing $N_{holes}$. These results implied that the degradation of the shelterbelts was caused by the beetle. We also found size-dependent mortality, i.e., only a few larvae can cause the death of smaller trees, but not larger trees. Furthermore, evaluation of the degradation at the stand level ($N_{holes}$) using a logistic regression analysis revealed that the degradation at the stand level could be predicted by $N_{holes}$. Our findings can be used as a useful index marker for diagnosing white birch shelterbelts.

**Keywords:** insect pests; number of adult exit holes; tree vigor; size-dependent mortality; diagnosis

## 1. Introduction

The genus *Anoplophora* (Coleoptera: Cerambycidae) includes some of the most damaging wood-boring pests in the northern hemisphere [1–4]. Although *Anoplophora* species have significant preferences for particular host trees [5,6], they have a very wide host plant range that includes more than 100 tree species [7]. Larvae of *Anoplophora* bore tunnel in the sapwood of living trees and grow, which disrupts the tree's vascular tissues, and possibly leads to the death of the tree [1,3,8]. Although many researchers have focused on the ecology of *Anoplophora* from the viewpoints of dispersal behavior [9–13], host preference [5,14,15], and invasion process [16–18], information on the forest degradation process caused by the beetles is scarce.

The white-spotted longicorn beetle (*A. malasiaca*), which is native to Japan, has long been a notorious pest of fruit trees such as citrus (*Citrus* spp.: Rutaceae) and pear (*Pyrus* spp.: Rosaceae), and ornamental trees such as sycamore (*Platanus*: Platanaceae) and maple (*Acer* spp.: Sapindaceae) [19]. In silviculture, damage to sugi cedar (*Cryptomeria japonica* [L. f.] D. Don: Taxodiaceae) plantations by this beetle has been reported in western Japan [20,21]. Thus, monoculture causes the mass attack of this beetle. Recently, degradation of shelterbelts composed of Japanese white birch (*Betula platyphylla* Sukaczev var. *japonica* [Miq.] Hara: Betulaceae) has been observed in central Hokkaido, Japan [22]

(Figure 1a). Abundant holes with a diameter of approximately 1 cm are often found at the trunk base (up to approximately 0.5 m above ground; Figure 1b) and exposed roots (Figure 1c), even in healthy appearing shelterbelts [22]. Some holes were clogged by fresh frass. While the life cycle of this longicorn beetle is not well-known in Hokkaido, the beetle requires 1 or 2 years to complete its life cycle and adults emerge in June in Shizuoka, central Japan [8]. Large diversity in body length was also reported, i.e., min. 19—max. 35 mm in the male and min. 21—max. 40 mm in the female [23]. Although in Hokkaido, the other three major wood-boring pests, i.e., *Agrilus viridis* (Linn.) (Buprestidae), *Opostegoides minodensis* Kuroko (Opostegidae) and *Phytobia betulae* (Kangas) (Agromyzidae), which attack living birch wood, were also reported [24,25], the evidence (frass and exit holes) [1] observed in the degraded white birch stands indicated that the infestation is largely by the white-spotted longicorn beetle. However, birch species are not likely to be a favored host of *Anoplophora* compared to citrus and maple [4,6,7]. So far only one case of serious infestation of white birch (*B. platyphylla*) by Asian longhorned beetle (*A. glabripennis* [Motschulsky]) was reported in China [26]. In Japan, mass attacks on white birch had been reported sporadically, but that did not result in tree death [27–29]. Because mass mortality of white birch caused by white-spotted longicorn beetle had not been reported in Japan, we wanted to clarify the effects of infestation on the degradation of white birch shelterbelts.

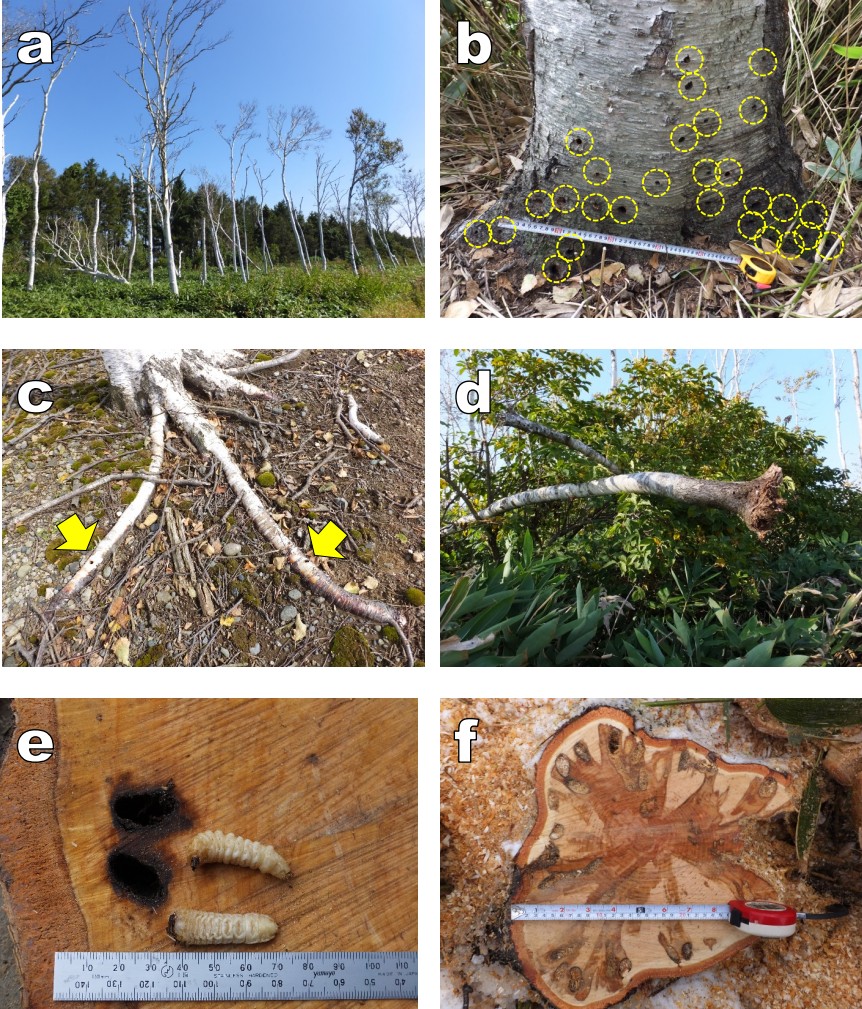

**Figure 1.** Infestation of white birch shelterbelts by white-spotted longicorn beetle. The following locations and plot numbers are shown in Figure 2. (**a**) Degraded stand at Plot 7 in Bibai on 21 September 2016. (**b**) Adult exit holes observed at the trunk base (dotted circles) at Plot 15 near the degraded stands, i.e., Plots 12 and 13. (**c**) Adult exit holes on the root, as shown by the arrows on 14

October 2016 (from [22]). (**d**) Snapping of trees at the trunk base at Plot 13 on 26 September 2016 (from [22]). (**e**) Larvae of white-spotted longicorn beetle in sapwood on 5 December 2016. (**f**) Cross-section of wood at ground level. Palmate discolored area elongates toward the tunnel bored by the larvae on 1 December 2016.

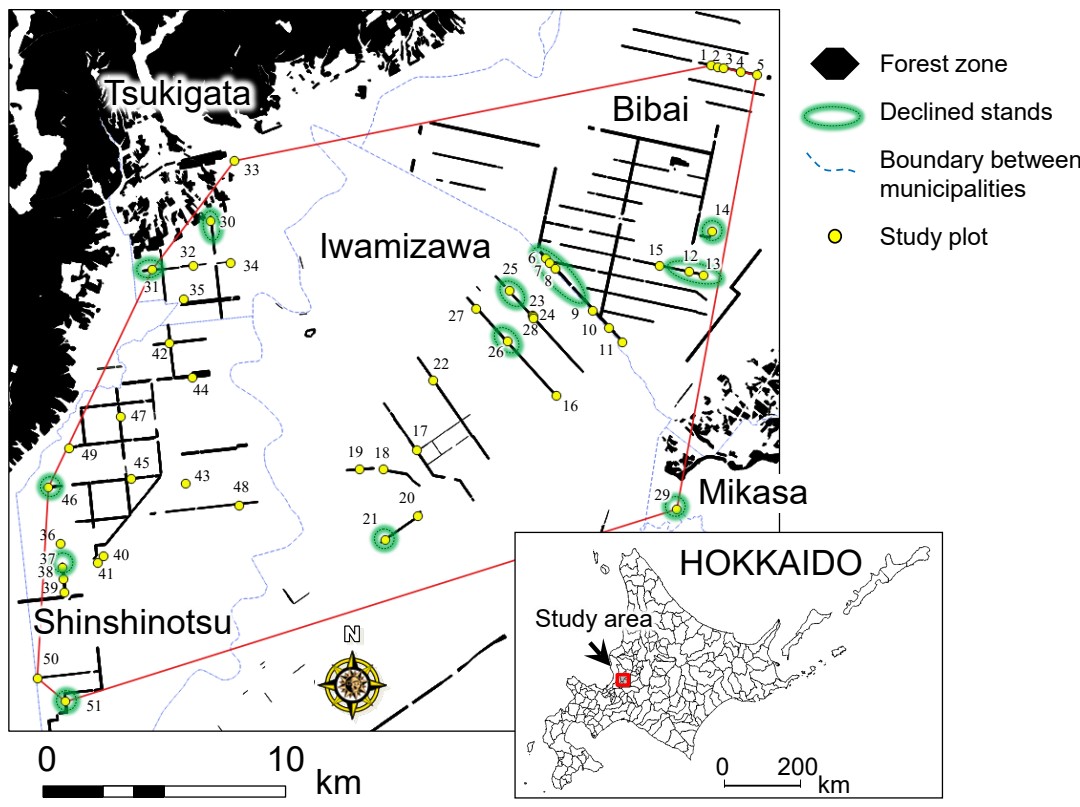

**Figure 2.** Study area, indicated by the polygon.

Standing dead trees of white birch are often observed in the old-growth stands [30]. The dead trees lose their shoots, twigs, and eventually large branches in the crown. Hence, only the trunks eventually stay, which then decay over time. However, many fallen trees were found in the degraded stands, and some of these trees still had leaves on twigs [22]. These fallen trees snapped at the trunk base without being uprooted (Figure 1d), which indicates that the trunk wood had been damaged severely. Larvae of the white-spotted longicorn beetle tunnel into the wood, which is manifested by a 'Swiss cheese' appearance [22] (Figure 1e,f). The porous wood is broken easily, and the tree consequently falls. Not only does the falling tree injure passers by on the farm road along the shelterbelts, but also fallen logs on farmland cause significant trouble to the farmers. As there are many white birch shelterbelts in central Hokkaido [31], management decisions about whether the stand should be removed at once must be made quickly. Therefore, we need to assess the extent of the damage to white birch shelterbelts caused by the longicorn beetle and demonstrate the risk of falling trees with respect to the severity of infestation. From the viewpoint of forest management, rather than pest control, it is worth verifying whether the number of adult exit holes can be used as an index of the decline of trees in shelterbelts.

We propose two assumptions to evaluate the effects of infestation, i.e., carrying capacity and lethal threshold in relation to tree size. First, the carrying capacity of each tree to hold larvae that can pupate and develop into adults is size-dependent. This means that there is an upper limit in wood volume as a food resource and living space for the larvae. Consequently, adult emergence from a tree increases with increasing tree size. Second, tree decline is affected by the severity of infestation and the tree size. For exam-

ple, only a few larvae can cause the death of smaller trees, but this is not the case for larger trees. This leads us to expect that there should be a correlation between the existence of a lethal threshold in the number of adult exit holes and the tree size. If so, the effects of infestation on tree vigor will differ according to tree size. In other words, the relationship between the number of adult exit holes and trees size should differ according to tree vigor.

In the present study, we addressed three questions: (1) To what extent does the number of adult exit holes correlated with the decline of white birch at the individual level? (2) Is there any relationship between the tree size and the minimum number of adult exit holes in the standing dead trees? and (3) To what extent does the severity of attacks correlate with the level of stand degradation?

## 2. Materials and Methods

### 2.1. Species Status

Although *A. malasiaca* is distributed in the four main islands of Japan; i.e., Hokkaido, Honshu, Kyushu, and Shikoku, and *A. chniensis* (Foster) inhabits China [32], European and American researchers consider that *A. malasiaca* is a synonym of *A. chinensis* [1]. However, *A. malasiaca* and *A. chniensis* can be discriminated by morphological characters [32] and DNA analysis [33,34]. Therefore, we used *A. malasiaca* as the scientific name in this study, following the Dictionary of Japanese Insect Names (Available online: http://konchudb.agr.agr.kyushu-u.ac.jp/index-j.html (accessed on 1 January 2020)).

### 2.2. Study Area

We carried out the investigation in Bibai, Mikasa, Iwamizawa, Tsukigata, and Shinshinotsu in central Hokkaido, in which degraded shelterbelts were observed (Figure 2). Shelterbelts in central Hokkaido were established after World War II on drained wetlands around the Ishikari River system [35], which has cool and humid conditions. The main purpose of establishing shelterbelts and windbreaks is to promote the growth of crops by increasing the soil temperature [36]. In addition to white birch, Manchurian ash (*Fraxinus mandshurica* Rupr. var. *japonica* Maxim.: Oleaceae), Dahurian larch (*Larix gmelinii* [Rupr.] Kuzen. ver. *japonica* [Maxim. ex Regel] Pilg.: Pinaceae), and Norway spruce (*Picea abies* [L.] H. Karst.: Pinaceae) have been used for the establishment of shelterbelts [31]. Each shelterbelt is composed of either a single or a combination of two birch species. Each has a length of approximately 500–3000 m and a width of approximately 20–50 m. The ground vegetation is often dominated by dwarf bamboo (*Sasa senanensis* [Fr. and Sav.] Rehder: Poaceae). 'Windbreaks' consisting of a few rows of trees (approximately 2–3 m in width) are also sporadically found in this region. The shelterbelts were established by the municipal or national forestry administrations, while the windbreaks were established by private organizations or farmers.

According to the Japan Meteorological Agency (1981–2010; Available online: http://www.data.kishou.go.jp/ (accessed on 25 September 2017)), annual precipitation in Bibai is 1156.5 mm year$^{-1}$, and the mean monthly temperatures during the warmest (August) and coldest (January) months are 26.4 °C and −12.0 °C, respectively. There is snow cover during November–April, with a mean annual maximum depth of 116 cm.

### 2.3. Relationship between the Number of Adult Exit Holes and Tree Vigor

We conducted the investigation in Bibai, Mikasa, Iwamizawa, Tsukigata, and Shinshinotsu in central Hokkaido, where degraded shelterbelts were observed (Figure 2).

In August–October 2015, we established 15 study plots in shelterbelts in Bibai (Figure 2). Plot size varied depending on the width and the growth stage of the shelterbelt (min. 0.007 ha to max. 0.056 ha). The average stand age was 59.5 (max. 62 to min. 55) years in 2015. Overall, six of the 15 plots were set up at degraded parts of shelterbelts

(Plots 6, 7, 8, 12, 13, 14), whereas Plots 9–11 and 15 were set up at the healthy-appearing parts of shelterbelts. Plots 1–5 were set up at the non-degraded shelterbelt far from the degraded stands; no white birch shelterbelts were observed from there to the northern boundary of Bibai. The plots in the non-degraded part were arranged at approximate even intervals in each shelterbelt. The degraded stands could often be distinguished from a distance because of the thin crowns [1], and many standing dead trees (Figure 1a). However, it was difficult to use one criterion to define "degraded stand" quantitatively, because there were many fallen trees instead of standing dead trees in some stands. Therefore, it was defined by the following criteria, according to the occasion: living trees were less than 40% of the expected tree density concerning the stand age (see Appendix A), or the percentage of standing dead trees was more than 30% of all standing trees. We counted the number of adult exit holes ($N_{holes}$) at the trunk base (up to approximately 0.5 m above ground), exposed roots, and also measured the trunk diameter at breast height (1.3 m height above ground, *DBH*; cm) of all living and standing dead trees in each plot. The crown vigor of each tree in the plots (hereafter *Vigor*; no unit) was assessed visually according to the following scale [37]: (0) no foliage in the crown, only a standing trunk remained, which was considered dead (standing dead tree); (1) less than 1/10 of the living foliage remained in the crown; (2) 1/10–1/4 of the foliage remained; (3) 1/4–1/2 of the foliage remained; (4) 1/2–3/4 of the foliage remained; (5) more than 3/4 of the foliage remained (intact crown). The abundance of dead branches in the crown helped to assess the crown vigor. The difference of frequency distribution of $N_{holes}$ among the degree of tree vigor (*Vigor*) was compared using generalized linear model (GLM). A Poisson distribution was used to measure the probability distribution of $N_{holes}$. To assess the influence of tree vigor on the relationship between $N_{holes}$ and *DBH*, we used generalized linear mixed model (GLMM) analyses as follows:

$$N_{holes} \sim \text{Poisson}(\mu),$$
$$\mu = \exp(a_{0j} + a_{1j}DBH) \tag{1}$$

This is the base model at the individual level. The model assumed that both the intercept ($a_{0j}$) and slope ($a_{1j}$) of the $N_{holes}$–*DBH* relationship in the *j*-th plot (*j* = 1 to 51) are associated with the degree of tree vigor (*Vigor*) as follows:

$$a_{ij} = b_{i0} + b_{i1}X + Plot_{ij}, \tag{2}$$

where $a_{ij}$ and $b_{ik}$ (*i* = 0, 1; *k* = 0, 1) are regression coefficients. *X* is the categorical variable, and *Vigor* is used in this analysis. Thus, the model represents the $N_{holes}$–*DBH* relationship with different *Vigor*s. $Plot_{ij}$ (*i* = 0, 1) is the random effect specific to each plot, in which $Plot_{0j}$ indicates the random intercept and $Plot_{1j}$ indicates the random slope at the *j*-th plot. Although *Vigor* is an ordinal variable, we treat it as a numerical variable to simplify the interpretation of the results.

### 2.4. Observation of Infested Wood Inside

Twelve white birches that fell in the shelterbelt near the Plot 9 on 1 December 2016, were studied to observe the condition of the wood inside infected by the white-spotted longicorn beetle. Trees were cut at the ground level, and the trunk was cut at 10 cm intervals up to 50 cm above the ground.

### 2.5. Estimating the Value of Longicorn Severity of Attacks as an Indicator of Stand Degradation

Degradation of the stands was correlated with $N_{holes}$ of standing dead trees. $N_{holes}$ does not increase after the death of the host tree because the longicorn beetle only infests living trees. Thus, the $N_{holes}$ in dead trees is a representative index specific to each plot, i.e., it indicates the severity of infestation in each plot.

In addition to the 15 plots in Bibai outlined above, we set up 36 plots at shelterbelts (*n* = 30), windbreaks (*n* = 2), roadside trees (*n* = 2), and natural forests (*n* = 2) composed

of white birch in Iwamizawa, Mikasa, Shinshinotsu, and Tsukigata (Figure 2) in October 2016 and April–June 2017. The average stand age of wind shelterbelts in 2015 was 42.7 (max. 62 to min. 23) years. We selected all degraded stands at first (*n* = 10; Plots 21, 24, 25, 26, 29–31, 37, 46, 51; Figure 3) and then set up plots at healthy-appearing stands to distribute stands evenly across the region. Note that Plot 21 should be classified as a degraded stand because most living trees in the plot had few leaves on the crown. As a reference, four stands were classified as 'nearly degraded' (Plot 9, 32, 38, 41) because of many dead canopy trees or fallen trees in addition to many exit holes. We counted $N_{holes}$ and measured the *DBH* of all living and standing dead trees in each plot. The difference in the $N_{holes}$–*DBH* relationship between living (*Alive*) and standing dead trees (*Dead*) was evaluated by Equations (1) and (2). Tree status (alive or dead) was the categorical variable in Equation (2).

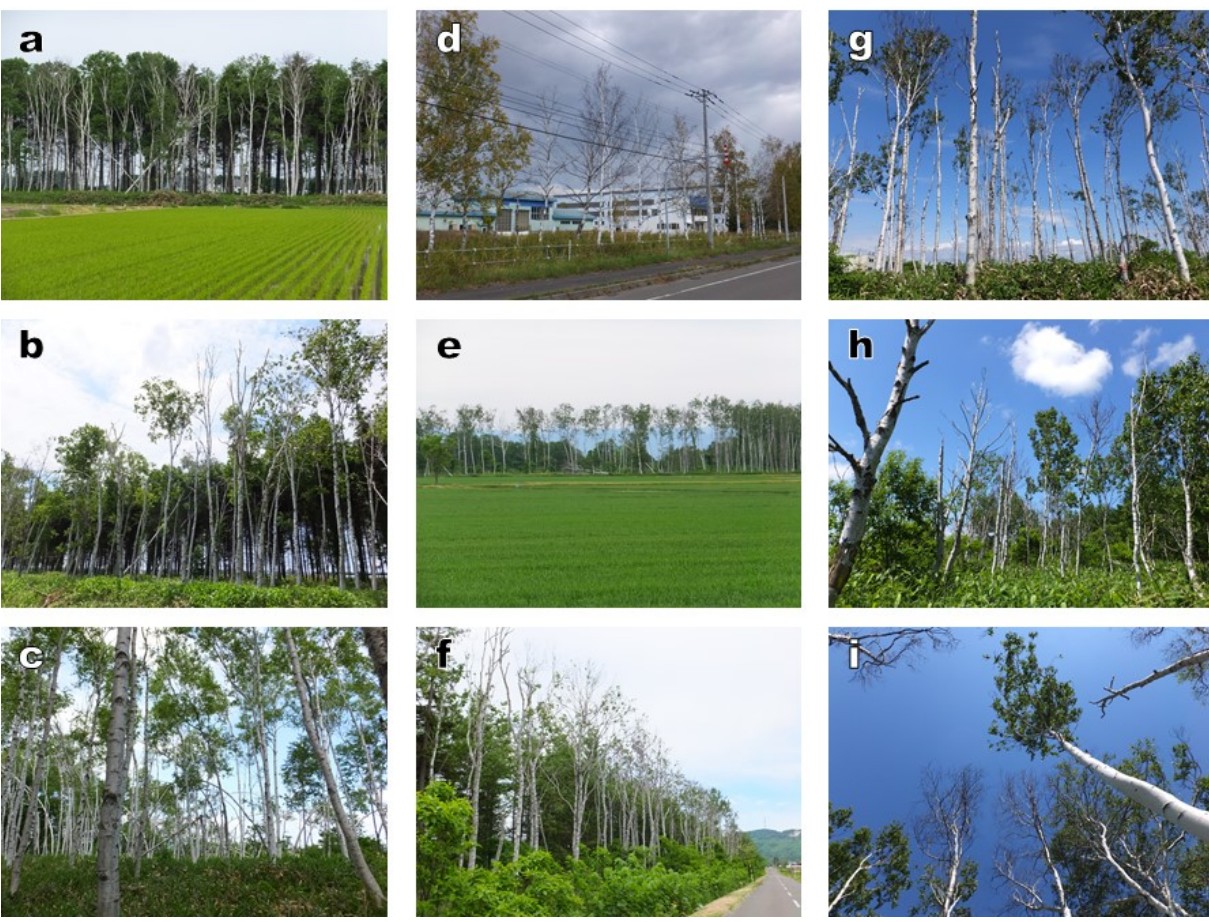

**Figure 3.** Study plots of the degraded stands in Iwamizawa, Mikasa, Shinshinotsu, and Tsukigata. (**a**) Plot 24 on 15 June 2017, (**b**) Plot 25 on 16 June 2017, (**c**) Plot 26 on 16 June 2017, (**d**) Plot 29 on 16 October 2016, (**e**) Plot 30 on 7 June 2017, (**f**) Plot 31 on 7 June 2017, (**g**) Plot 37 on 16 June 2017, (**h**) Plot 46 on 13 June 2017, (**i**) Plot 51 on 14 June 2017. Note that we have no photos of Plot 21 in the growing season because the investigation was carried out on 29 March 2017. Crown vigor in Plot 21 was observed during the growing season in 2016.

The probability that a stand becomes degraded as a function of $N_{holes}$ was evaluated by logistic regression analyses (degraded stand = 1, healthy appearing- and nearly degraded stand = 0). Because the size distribution of trees differed among plots due to different stand ages, we proposed a criterion to compare the severity of infestation among plots. We estimated the average $N_{holes}$ for standing dead trees with a *DBH* of 25 cm (hereafter referred to as $N_{D25}$) in each plot using Equation (1). The $N_{D25}$ of standing dead trees in each plot was obtained by substituting *DBH* = 25 cm into the model.

### 2.6. Statistical Analyses

We used R ver. 4.0.2 [38] for GLM and GLMM analyses, with the *glmmTMB* function in the package *glmmTMB* to pay special attention to the zero-inflated probability. Coefficients of the random intercept and random slope for each plot were output by the *ranef* function. Logistic regression analyses were conducted using the *glm* function with the logit link, as the binomial distribution was assumed for the objective variable. Akaike's information criterion (AIC), which balances the fit of the model against the number of parameters, was used to select the best-fit model for the GLM and GLMM [39]. The model with the smallest AIC value was accepted as the best-fit model for the data [40,41].

## 3. Results

### 3.1. Infected Wood Inside

In our examination of wood cross-sections, we often observed palmate-discolored areas (Figures 1e,f and 4) [42]. The tunnels were often clogged by rotten and blackened sawdust, which could be related to infection by wood rot fungi. The tunnels bored by the larvae were not observed at >40 cm in height (Figure 4).

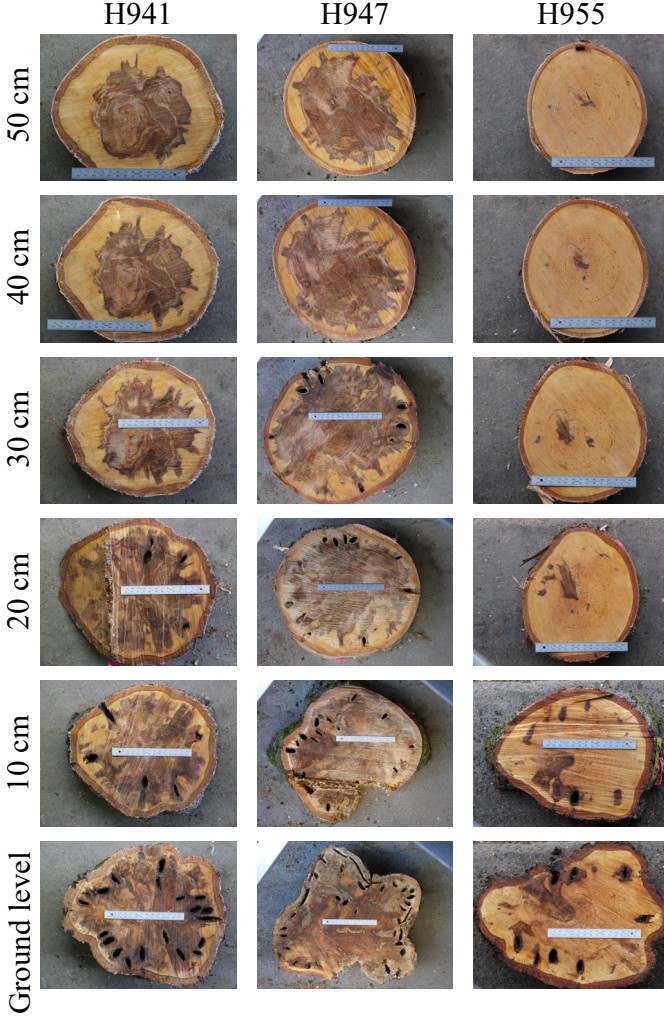

**Figure 4.** Examples of cross-sections of infested trees (#H941, #H947, and #H955) with different heights above the ground. Wood was decayed severely in #H941 and #H947, but was almost intact in #H955. Scale = 15 cm.

*3.2. Relationship between the Decline of White Birch and Infestation by Longicorn Beetle*

The frequency distribution of $N_{holes}$ differed markedly among tree vigor, and $N_{holes}$ of standing dead trees (*Vigor* = 0) was greater than that of living trees (*Vigor* = 1–5) (Figure 5a; AIC = 5731.22 for the best model and AIC = 7392.65 for the null model). The $N_{holes}$–*DBH* relationship for living and dead white birch is shown in Figure 5b. As expected, $N_{holes}$ increased with increasing *DBH*. However, the tendency differed between living and dead trees. In the GLMM analyses, both *DBH* and *Vigor* were included in the best model (Table 1). *Vigor* reflected the intercept in the $N_{holes}$–*DBH* relationship, with the negative value of the coefficient indicating that the intercept increased with decreasing tree vigor. The odds ratio suggested that $N_{holes}$ of dead trees (*Vigor* = 0) tended to be approximately twice (=$e^{-0.155 \times 0}/e^{-0.155 \times 5}$) as large as that of intact trees (*Vigor* = 5). These results imply that $N_{holes}$ increases with declining tree vigor.

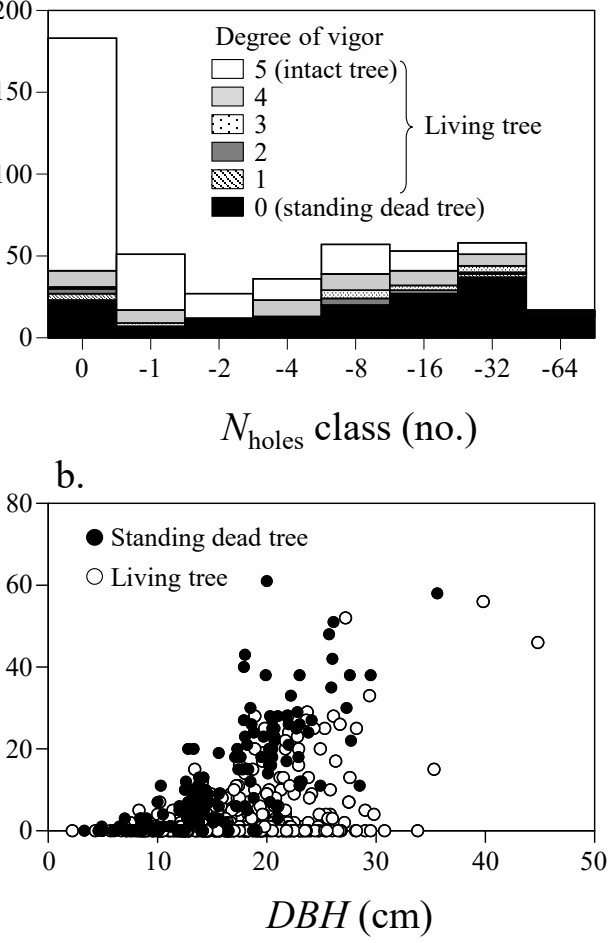

**Figure 5.** Difference in the number of adult exit holes ($N_{holes}$) among living trees and standing dead trees. (**a**) Log-scaled $N_{holes}$-class frequency histogram of the number of individuals and (**b**) the relationship between $N_{holes}$ and *DBH*. Living trees are shown by single symbol regardless of the degree of vigor (*n* = 352 for living trees, *n* = 143 for dead trees).

**Table 1.** The best model for the number of adult exit holes ($N_{holes}$) in relation to *DBH* and tree vigor (*Vigor*). *Coeff.*; estimated coefficient, *SE*; standard error.

| Variables | Coeff. | SE | z |
|---|---|---|---|
| Intercept | −0.506 | 0.527 | −0.960 |
| *DBH* | 0.094 | 0.016 | 5.969 |
| *Vigor* | −0.155 | 0.011 | −14.304 |

Notes: Variance and standard deviation (*SD*) of the random effect of the intercept and the slope was 2.997 (*SD* = 1.73) and 0.00079 (*SD* = 0.028), respectively. The estimated intercept of the zero-inflation model was −2.054 (*SE* = 0.236, *z* = −8.69).

A significant relationship was also found between $N_{holes}$ and corresponding dead trees with a minimum *DBH* ($r^2$ = 0.617, *p* < 0.001; Figure 6). Thus, $N_{holes}$ of small dead trees were less than those of large dead trees. This result implies the existence of a size-dependent threshold for $N_{holes}$ for dead or alive white birch, i.e., the tree will soon die if $N_{holes}$ exceeds the threshold.

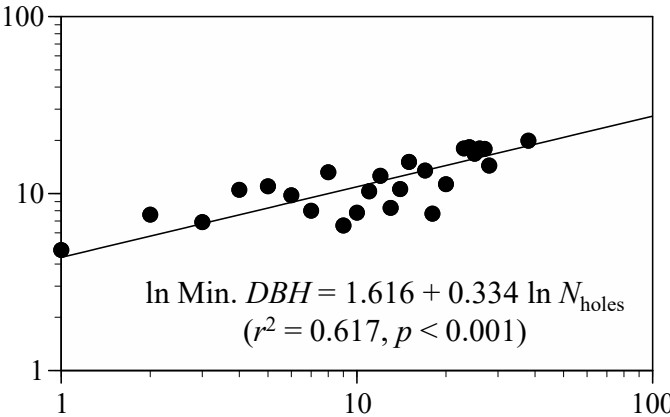

Number of adult exit holes, $N_{holes}$ (no.)

**Figure 6.** Relationship between the number of adult exit holes ($N_{holes}$) and the corresponding minimum *DBH* of standing dead trees. We used the data from more than five individuals for each $N_{holes}$ to consider the deviation.

### 3.3. Estimating the Value of Longicorn Severity of Attacks as an Indicator of Stand Degradation

Degraded stands were found irrespective of the year of establishment (Figure 7). The frequency distribution of $N_{D25}$ of standing dead tree was shown in Figure 8a. Degraded stands occurred when $N_{D25}$ exceeded 16 (=$2^4$). If $N_{D25}$ exceeded 32 (=$2^5$), almost all stands were degraded. The probability of occurrence of degraded stands with respect to $N_{D25}$ could be estimated by the following logistic regression model (Table 2, Figure 8b):

$$Pr. = \frac{1}{1+\exp(4.155-0.156N_{D25})},$$ (3)

in which the AIC of the model was 39.202 (residual deviance [r.d.] = 35.202, degrees of freedom [d.f.] = 49), and that of the null model was 63.449. A 50% reaction could be estimated from Equation (3), and we found that a stand has a 50% probability of degradation at $N_{D25}$ = 26.7. Note that the largest $N_{D25}$ (=96.0) observed in Plot 46 seems to be an outlier (Figure 8b), and the value may influence the result. However, nearly the same result was obtained, even though Plot 46 was excluded from the analysis (AIC = 39.202, r.d. = 35.202, d.f. = 48).

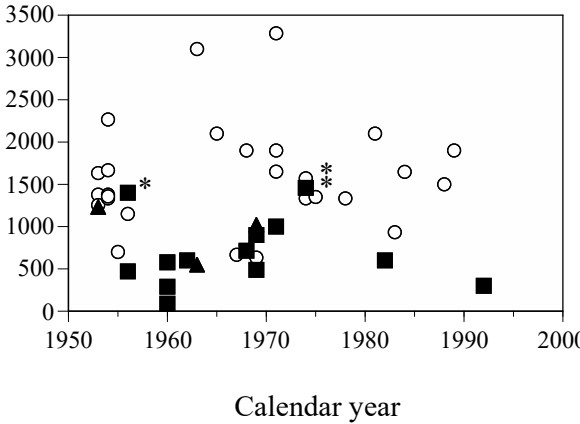

Calendar year

**Figure 7.** Stand density of each study plot with respect to year of establishment. ○, healthy appearing stand; ▲, nearly degraded stand; ■, degraded stand. *, Plot 12 was composed of slender trees that would be associated with relatively high density. ⚬̇, Plot 21 in which there were many trees with little foliage in the crown.

**Table 2.** The best model for the number of adult exit holes ($N_{holes}$) corresponding to *DBH* and *survival* (*dead* or *alive*). N = 1246 for living trees and n = 338 for dead trees. *Coeff.*; estimated coefficient, *SE*; standard error.

| Variables | Coeff. | SE | z |
|---|---|---|---|
| *Intercept* | −0.831 | 0.288 | −2.890 |
| *Survival_dead* | 0.395 | 0.102 | 3.891 |
| *DBH × Survival_alive* | 0.088 | 0.008 | 10.816 |

Notes: *Survival_alive* is the reference category (=0). Variance and standard deviation (SD) of the random effect of the intercept and the slope was 3.253 (SD = 1.803) and 0.015 (SD = 0.038), respectively. The estimated intercept of the zero-inflation model was −1.751 (*SE* = 0.112, *z* = −15.650).

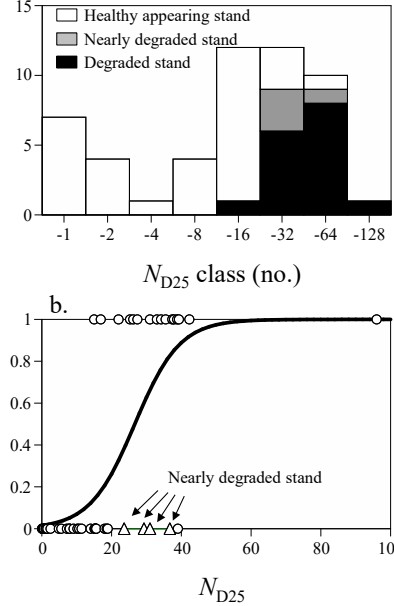

**Figure 8.** Appearance of degraded stands. (**a**) Log-scaled $N_{D25}$-class frequency histogram of the number of stands and (**b**) the logistic regression curve regarding the probability of appearance of degraded stand in correlation to $N_{D25}$. The nearly degraded stands (∆) were included as the healthy appearing stands in the analysis.

## 4. Discussion

### 4.1. Infected Wood Inside

The number of adult exit holes of the white-spotted longicorn beetle ($N_{holes}$) in dead white birch trees tends to be greater than that in living ones, and a negative relationship between crown vigor and $N_{holes}$ was found, suggesting that $N_{holes}$ is a good indicator of tree vigor. Furthermore, the probability of occurrence of the stand degradation increased with increasing $N_{holes}$ (Figure 8). These results show that the level of attacks by the white-spotted longicorn beetle is a good indicator of the degradation of white birch shelterbelts in central Hokkaido. However, for small trees, death was observed without adult exit holes (Figure 5a). The mortality of small trees, in general, was caused by shading. As white birch is a pioneer species, suppression often causes mortality of small trees of this species.

Larvae of the white-spotted longicorn beetles bore tunnels in the cambial region and sapwood (Figures 1e,f, and 4) [1]. After the repeated attack, the larval galleries disrupt the tree's vascular tissues, which possibly leads to the death of the tree [1]. We observed the discolored area on almost the entire surface of the cross section of the wood #H941 and #H947 (Figure 4). Generally, the living sapwood is resistant to decay compared with the heart wood [43], and the heart rot is often observed in the old white birch [44,45]. Therefore, the sap rot observed in #H941 and #H947 implies the infestation by the white-spotted longicorn beetles (also see Figure 1f). Moreover, clogging by rotten and blackened sawdust in the tunnel can be a breeding ground for wood rot fungi. It is probable that the sap rot accelerates tree death. It is considered that crown dieback reflects the severity of attack in relation to the degree of disruption of the cambial region. In addition, we have to consider the possibility that some larvae that did not pupate also bored tunnels in the wood but did not produce exit holes. Although at present there is no information about the mortality of the larvae in white birch, it was documented that the mortality rate of eggs in citrus trees decreased with the increase of the number of exit holes [46]. The number of adult exit holes surveyed in this study might underestimate the severity of infestation.

The tolerance of trees to beetle attacks must be size-dependent because the cambial region is proportional to *DBH* to a large extent. Indeed, a positive relationship between the number of adult exit holes of white-spotted longicorn beetle and tree diameter was reported in citrus [46]. The plausible assumption is that the maximum cumulative capacity of larvae in each tree is also size-dependent, which allows us to consider the size-dependent lethal threshold of $N_{holes}$. Since the regression between $N_{holes}$ and the corresponding minimum *DBH* of standing dead trees shows the critical stage of infestation, we may use it as a simple diagnostic criterion for on-site decisions, i.e., if $N_{holes}$ of a tree reaches the line, the tree would likely die soon.

### 4.2. Degradation of White Birch Shelterbelts

Severe beetle infestation may lead to the degradation of white birch shelterbelts. A degraded stand occurred if $N_{D25}$ exceeds 16 (Figure 8a). The logistic regression curve suggests that a stand has a 50% probability of degradation if $N_{D25}$ exceeds 25 (Figure 8b). These results imply that the destruction of the white birch stands linked with the white-spotted longicorn beetle progresses rapidly. This is consistent with the finding that there are many larvae of white-spotted longicorn beetles in the trees at stands, where degradation is progressing. Trees should be removed as early as possible if $N_{D25}$ exceeds 16. The infested wood must be burned or fumigated after cutting to inhibit further emergence of adults, as pupae may still exist in the wood.

### 4.3. Cause of Massive Mortality

It is currently unclear why the massive mortality of the white birch linked with the white-spotted longicorn beetle has occurred in central Hokkaido. The year of the establishment was not strongly correlated with degradation (Figure 7), which implies that the

degradation occurred recently. Generally, stressed trees are more susceptible to attacks by insects than their healthy counterparts [47]. Since white birch shows little tolerance to flooding [48], the trees in the wind-shelter belts might be stressed with age due to relatively high ground water level in the drained peatland [49]. This may increase the susceptibility to white-spotted longicorn beetle. Monoculture may also be a plausible factor to explain the massive mortality of white birch. Monocultures are more susceptible to pests because they can provide a huge food source and ideal habitat for pest insects [50]. For example, in Japan, pine forests and oak forests were destroyed by the pine-wilt disease and the oak wilt disease, respectively, for the last several decades [51,52]. Black pine (*Pinus thunbergii* Parlat.: Pinaceae) has been used to establish the coastal forest to prevent sand movement [53], and red pine (*P. densiflora* Sieb. *et* Zucc.) has often regenerated in the area devastated by overcutting and fire [54]. The wood of konara oak (*Quercus serrata* Thunb.: Fagaceae) has been used for fuel, and the oak forests have developed around the villages due to the coppicing [55]. Historical land use led to the dominance of these tree species, and consequently, they provide a bountiful food source for the pests. Similarly, the establishment of white birch shelterbelts provides a preferable habitat for the white-spotted longicorn beetle.

*4.4. Pest Control for the Shelterbelts*

A few effective pest control techniques have been developed for *Anoplophora* on any of their hosts [1]. Because the larvae of the white-spotted longicorn beetle bore into trunks, they are difficult to control, and damage to the hosts is often difficult to detect [14]. Biological control using fiber bands containing the entomopathogenic fungi against the longicorn beetle has already been developed [1]. This method may be effective for roadside trees in urban areas, but it seems unrealistic for shelterbelts because of the enormous number of trees. Hence, it may be realistic to moving away from monoculture by exchanging white birch for other species. Since mixed-species plantations have a higher advantage than monocultures in pest control [50], shelterbelts composed of mixed-species can minimize the risk of the degradation caused by pests.

**5. Conclusions**

Our study focused on the number of adult exit holes as an index of the severity of infestation. The approach developed in this study allowed us to show that the level of attacks by the white-spotted longicorn beetle is a good indicator of the degree of decline of birch trees used in shelterbelts. Whether the beetle is responsible for tree death is possible but should be further investigated. There is no information on how far the beetle can move in the agricultural landscape in central Hokkaido, but the scattering of degraded shelterbelts in our study area (Figure 2) implies that white-spotted longicorn beetle attacks can occur anywhere. Indeed, degraded roadside white birch trees were found at an avenue in Sapporo in 2017, about 50 km from Bibai. Even if we find some adult exit holes on the trunk, it will be difficult to remove the tree immediately because of no risk measurement. Visual inspections based on the number of adult exit holes (Figures 4 and 8) together with symptoms such as a thin crown [1] will contribute to the diagnosis of infested white birch.

**Author Contributions:** Conceptualization, writing, methodology, K.M.; investigation, all authors; review of wind shelterbelt, K.I.; ecological implication of the degradation by the longicorn beetle, M.H.; wood detection, Y.W.; review and editing, all authors. All authors have read and agreed to the published version of the manuscript.

**Funding:** This research received no external funding.

**Data Availability Statement:** Not applicable.

**Acknowledgments:** We are grateful to Sorachi General Sub prefectural Bureau and Bibai City Office for permission to cut the trees in the regally protected wind-shelter belt; to T. Abe, Y. Koba-

yashi, A. Kokubo, K. Hayasaka, R. Nagasawa, S. Nishimura, I. Tanahashi for their help in field investigation (in alphabetical order). We also thank Abidur Rahman, Faculty of Agriculture, Iwate University, and the Academic Editor of the journal for critical reading of the manuscript and editing.

**Conflicts of Interest:** The authors declare no conflict of interest.

## Appendix A

The relationship between the tree volume ($V$; m$^3$/ha) and density ($N$; no./ha) in the stand is fitted by the following reciprocal equation:

$$\frac{1}{V} = A + \frac{B}{N'} \tag{A1}$$

in which $A$ and $B$ are parameters specific to the stand age and the height of the stand [56]. In white birch stand, both $A$ and $B$ are a function of the stand height ($Hd$) as follows [57]:

$$A = 7.722 \times 10^{-2}\, e^{-0.205Hd} + 2.112 \times 10^{-3}, \tag{A2}$$

$$B = 8.653 \times 10^{4}\, Hd^{-3.780}, \tag{A3}$$

Self-thinning curve on the density-control diagram for white birch is described by the following equation:

$$V = 9.90 \times 10^{4} N^{-0.734} \left[1 - \left(\frac{N}{N_0}\right)^{1.734}\right], \tag{A4}$$

in which $N_0$ (no./ha) indicates the initial planting density [57]. In Hokkaido, the planting density of a wind shelterbelt is 5000 saplings per hectare.

On the other hand, the height growth curve of the tree is strongly influenced by the site quality (hereafter site index; $SI$). The $SI$ for the white birch was represented by the stand height at 50 years old, and the $Hd$ at stand age $t$ at a given $SI$ is described by

$$Hd(t) = f_1(t) + \frac{f_2(t)}{f_2(50)}[SI - f_1(50)], \tag{A5}$$

in which $f_1(t) = 22.2254 - 23.8003 \times 0.9692^t$ and $f_2(t) = 3.6484 - 3.1549 \times 0.96^t$ [58]. As we had no information about the stand height of each study plot in this study, we used $SI = 18$ for convenience [59]. From Equations (A1)–(A4), we can estimate the hypothetical stand density ($N$) of each plot at stand age $t$ using numerical analyses.

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
