# Peer review of "Degradation of White Birch Shelterbelts by the Attack of White-Spotted Longicorn Beetles in Central Hokkaido, Northern Japan"

_forests, doi:10.3390/f13010034_

Round 1

Reviewer 1 Report

Dear Authors,

I heave only one comment to this interesting and well-prepared manuscript. In recent decades, invasions of some native as well as alien beetle species have been observed in many forested areas of the world. Some of these observations appear to be closely related to global climate changes. Unfortunately, in the reviewed article I cannot find in the paper any assumptions as to the causes of the observed attack of the aforementioned longicorn beetle on Hokkaido (?).

With best wishes,

Adrian Smolis 

Author Response

Thank you for your revision. As we stated in our response to your comments, our study focused on the number of adult exit holes as an index of the severity of infestation for the forest management rather than pest control. It is definitely interesting to demonstrate the effect of global climate change on the risk of the outbreak of longicorn beetle. Thank you.

Warm regards,

Authors

[Comment] I heave only one comment to this interesting and well-prepared manuscript. In recent decades, invasions of some native as well as alien beetle species have been observed in many forested areas of the world. Some of these observations appear to be closely related to global climate changes. Unfortunately, in the reviewed article I cannot find in the paper any assumptions as to the causes of the observed attack of the aforementioned longicorn beetle on Hokkaido (?).

[Response] Thank you for your invaluable suggestion about the relationship between the outbreak of the longicorn beetle and the global climate change. The global warming may contribute to the outbreaks, and we have to pay attention to the effects of the global warming on the growth and the reproduction of the longicorn beetle. But we have no information about the temperature-dependent growth model of this beetle species in Hokkaido. Because we are not entomologists and the aim of this study is to evaluate the number of adult exit holes as an index of the severity of infestation for the forest management rather than pest control. We consider that the effects of the global warming should be evaluated in the further studies. Thank you!

P.S. Our English in this revised draft was revised by Prof. Rahman at Iwate University.

Reviewer 2 Report

This study research on the key elements for diagnosing white birch shelterbelts, and implied that the degradation of the shelterbelts was caused by Anoplophora malasiaca. The topic of this manuscript is very important, in general, the manuscript is well-organized, with the clear statement of objectives to guide the organization of the document. The language used in some part of this manuscript is not easy to follow for the general audience, it will really benefit/improved if an input from a Native English speaker is incorporated. In addition to this, the following points summarize major concerns that I have noted:

L108: At least one or two figures showing the key morphological features should be added, indicating the key differences with A. chinensis

Line 146: To evaluate the relationship between the number of adult exit holes and tree vigor, you should determine white-spotted longicorn beetle is the main cause of degradation of shelterbelts. Please specify.

Line 147: What healthy appearing parts mean? According to Fig.1b, no obvious crown vigor change with many adult exit holes observed at the trunk base?

Lines 157-158: Please specify the method for counting numbers of adult exit holes at the tree level.

Line 160: Example of typical crown of different degree of tree vigor are better to be presented as figures.

Line 167: Why you choose GLMM for analysis? Please specify.

Lines 189-190: Why did you set up additional 33 plots under different stand conditions? Please specify.

Line 201: Please specify Eqs. (1) and (2).

Line 242-243: Please do a difference analysis to prove your conclusion rather a simple bar chart.

Line 244: To make the result obvious, please remake the Fig.5b at DBH 0-20cm.

Line 371: That's an incomplete statement,as far as I know, there are several effective pest control techniques for Anoplophora, especially the lure host trees and reasonable combination of tree species in forest management. Natural enemy species are also  

In the abstract, 51 stands were chosen for study as you said. But in the methods, you just describe you chose 48 plots. Please specify sample stands selection.

In this study, many of the results are just simple statistics and not well analyzed, and too focused on experimental modelling, rather than current pest management.

This experiment does not explore the relationship between plots of different degradation degree. Please simplify the introduction of plots chosen in Materials and Methods.

Author Response

Thank you for your invaluable comments. We revised our manuscript to follow the comments. But we could not follow some of them, because of our lack of skill. Please accept our apology. Revised parts in our text were colored by yellow marker. Thank you.

Warm regards,

Authors

This study research on the key elements for diagnosing white birch shelterbelts, and implied that the degradation of the shelterbelts was caused by Anoplophora malasiaca. The topic of this manuscript is very important, in general, the manuscript is well-organized, with the clear statement of objectives to guide the organization of the document. The language used in some part of this manuscript is not easy to follow for the general audience, it will really benefit/improved if an input from a Native English speaker is incorporated. In addition to this, the following points summarize major concerns that I have noted:

[Response] Before submission, our English was corrected by Textcheck, an English consultant (http://www.textcheck.com/jp/text/page/index). Our English in the revised draft was also revised by Prof. Rahman at Iwate University. Corrected parts were not colored by yellow. Thank you.

L108: At least one or two figures showing the key morphological features should be added, indicating the key differences with A. chinensis

[Response] We added the explanation regarding the morphological difference between two Anoplophora species, but we have no figures about it. Because we are not entomologists, all we can show the information about the difference between two Anoplophora species is just ‘2.1. Species status’.

Line 146: To evaluate the relationship between the number of adult exit holes and tree vigor, you should determine white-spotted longicorn beetle is the main cause of degradation of shelterbelts. Please specify.

[Response] To determine the cause of degradation of shelterbelts, we have to evaluate the relationship between the number of adult exit holes and tree vigor in this study. We ask for your kind understanding.

Line 147: What healthy appearing parts mean? According to Fig.1b, no obvious crown vigor change with many adult exit holes observed at the trunk base?

[Response] The size of the tree in Fig.1b was the largest in this study, and therefore, we could not observe no obvious change in crown vigor. We consider that this tree did not reach the lethal threshold yet, and the stand was not degraded yet.

Lines 157-158: Please specify the method for counting numbers of adult exit holes at the tree level.

[Response] In lines 157-158, we explained two tasks (counting the number and measuring the trunk diameter). It might confuse the reviewer. We revised this part as follows:

Before: “We counted the number of adult exit holes (Nholes) and measured the diameter of…”

After: “We counted the number of adult exit holes (Nholes) at the trunk base (up to ca. 0.5 m above ground) and exposed roots, and measured the trunk diameter of…”

Line 160: Example of typical crown of different degree of tree vigor are better to be presented as figures.

[Response] We also consider that it will be better to show the typical crown of different degree of tree vigor. But we have good pictures to show the vigor of crown, because the crown often overlapped with the other crowns in the dense stand. Moreover, it will take much time to illustrate the crown vigor, and this is more than we can handle. We ask for your kind understanding.

Line 167: Why you choose GLMM for analysis? Please specify.

[Response] We added the explanation why we must use GLMM as follows:

Before: “we conducted generalized linear mixed model (GLMM) analyses as follows: “

After: “we conducted generalized linear mixed model (GLMM) analyses to understand the random effects specific to the stand as follows:”

Lines 189-190: Why did you set up additional 33 plots under different stand conditions? Please specify.

[Response] We noticed the degradation of wind shelterbelt at Bibai at first. In that time, we did not have good idea how we investigate and evaluate the degradation. It is because that it was the first attempt to investigate the degradation caused by the forest pest. Besides, we had enough time to investigate the degradation because we had other several studies in that time. After the first investigation, we found the degraded stands around our home town. We come to realize the seriousness of the matter. Then, we conducted the second investigation. But this is the inside story and we hesitate to specify why we conducted two assessments. 

Line 201: Please specify Eqs. (1) and (2).

[Response] We just used the Eqs. (1) and (2) for the analysis. Why we described the categorical variable as X is to avoid the duplication of the explanation. ‘Vigor’ was applied to X for the first investigation, and ‘dead or alive’ was applied to X for the all investigations. We ask for your kind understanding.

Line 242-243: Please do a difference analysis to prove your conclusion rather a simple bar chart.

[Response] We added the method and result of new analysis about the frequency distribution shown in Fig. 5a.

“The difference of frequency distribution of Nholes among the degree of tree vigor (Vigor) was compared by generalized linear model (GLM). A Poisson distribution was used to measure the probability distribution of Nholes.” was added before “To assess the influence of tree vigor on the relationship between Nholes and DBH, …” at L164.

We revised the L242 as follows:

Before: “…than that of living trees (Vigor = 1–5) (Figure 5a).”

After: “…than that of living trees (Vigor = 1–5) (Figure 5a; AIC = 5731.22 for the best model and AIC = 7392.65 for the null model).”

Line 244: To make the result obvious, please remake the Fig.5b at DBH 0-20cm.

[Response] There are so many data in this graph, and it is difficult to show the difference between standing dead trees and living trees at DBH 0-20cm. But the small trees are not important, because the mortality of the small trees will be caused by shading or only a few infestations. We consider that it is important to show the difference in the range of DBH = 15 <.

Line 371: That's an incomplete statement,as far as I know, there are several effective pest control techniques for Anoplophora, especially the lure host trees and reasonable combination of tree species in forest management. Natural enemy species are also  

[Response] We are sincerely sorry for the lack of our knowledge about the lure host tree. It was saying too much. But the aim of our study is just forest management rather than pest control. Then, we revised the head of the paragraph as follows:

Before: “No effective pest control technique has been developed…”

After: “A few effective pest control techniques have been developed…”

In the abstract, 51 stands were chosen for study as you said. But in the methods, you just describe you chose 48 plots. Please specify sample stands selection.

[Response] We are sorry for the careless mistake. ‘51 stands’ is correct.

In this study, many of the results are just simple statistics and not well analyzed, and too focused on experimental modelling, rather than current pest management.

[Response] Not only this study is our first attempt, but also we could not find the previous researches that focused on the number of adult exit holes as an index of the severity of infestation. We hope that our study serves as a test of the index.

This experiment does not explore the relationship between plots of different degradation degree. Please simplify the introduction of plots chosen in Materials and Methods.

[Response] As we responded to the comment on line 189-190, we had no other choice but to conduct two field investigations. There were ca. 350 alive trees (+ ca. 150 standing dead trees) in the first investigation, and therefore, we could conduct the observation of crown vigor. In the second investigation, there were ca. 900 alive trees (+ ca. 190 standing dead trees) and it was difficult to conduct the observation of crown vigor. All we could do was to record “dead or alive” in the second investigation within a limited time. This is the inside story and we ask for your kind understanding.

Round 2

Reviewer 2 Report

This work used a logistic regression analysis revealed that the tree degradation at the stand level could be predicted by the number of exit holes

(Nholes). This  can be an  useful index marker for diagnosing trees damaged by longhornbeetles. 

Still, there are some redundant expression and some grammar errors need to be check again and revise accordingly. 

Author Response

Dear Reviewer 2

We sincerely thank the reviewer for the positive comments. We have carefully read the manuscript and corrected the mistakes accordingly, which can be found in the track change.

We hope that this version of the manuscript will satisfy the reviewer and meet the publication standard of forests.

Warm regards,

Author and co-authors
